# Measuring Walking Speed Failed to Predict Early Death and Toxicity in Elderly Patients with Metastatic Non-Small-Cell Lung Cancer (NSCLC) Selected for Undergoing First-Line Systemic Treatment: An Observational Exploratory Study

**DOI:** 10.3390/cancers14051344

**Published:** 2022-03-05

**Authors:** Amélie Aregui, Johan Pluvy, Manuel Sanchez, Theresa Israel, Hélène Esnault, Alice Guyard, Marie Meyer, Antoine Khalil, Gérard Zalcman, Agathe Raynaud Simon, Valérie Gounant

**Affiliations:** 1Geriatrics Department, Bichat-Claude Bernard Hospital, Cancer Institute AP-HP.Nord, Université de Paris, 75018 Paris, France; amelie.aregui@aphp.fr (A.A.); manuel.sanchez@aphp.fr (M.S.); helene.esnault@aphp.fr (H.E.); agathe.raynaud-simon@aphp.fr (A.R.S.); 2Department of Thoracic Oncology, CIC-1425 Inserm, Bichat-Claude Bernard Hospital, Cancer Institute AP-HP.Nord, Université de Paris, 75018 Paris, France; johan.pluvy@ap-hm.fr (J.P.); gerard.zalcman@aphp.fr (G.Z.); 3Department of Medical Imaging, Bichat-Claude Bernard Hospital, Cancer Institute AP-HP.Nord, Université de Paris, 75018 Paris, France; theresa.israel@aphp.fr (T.I.); antoine.khalil@aphp.fr (A.K.); 4Department of Pathology, Bichat-Claude Bernard Hospital, Cancer Institute AP-HP.Nord, Université de Paris, 75018 Paris, France; alice.guyard@aphp.fr; 5Department of Dietetics, Bichat-Claude Bernard Hospital, Cancer Institute AP-HP.Nord, Université de Paris, 75018 Paris, France; marie.meyer@aphp.fr

**Keywords:** NSCLC, walking speed, older patients, toxicity, survival

## Abstract

**Simple Summary:**

Lung cancer is common in elderly adults. Onco-geriatric tools are meant to constitute a global approach designed to help oncologists to determine which elderly patients could benefit from systemic treatments, without major safety issues. This evaluation can prove to be time- and resource-consuming. The challenge is to find an easy and reproducible test, meant to guide the clinician’s decisions. Walking speed has emerged as a potential predictor of mortality in elderly cancer patients, yet data involving lung cancer patients are scarce. Our prospective exploratory study sought to determine whether walking speed would predict early death or toxicity in patients with metastatic lung cancer receiving first-line systemic intravenous treatment. Our results revealed walking speed to be numerically, yet not significantly, associated with early mortality in older metastatic lung cancer patients. Following these hypothesis-generating results, a larger prospective, multicenter study appears to be required to further investigate this outcome.

**Abstract:**

Walking speed (WS) has emerged as a potential predictor of mortality in elderly cancer patients, yet data involving non-small-cell lung cancer (NSCLC) patients are scarce. Our prospective exploratory study sought to determine whether WS would predict early death or toxicity in patients with advanced NSCLC receiving first-line systemic intravenous treatment. Overall, 145 patients of ≥70 years were diagnosed with NSCLC over 19 months, 91 of whom displayed locally-advanced or metastatic cancer. As first-line treatment, 21 (23%) patients received best supportive care, 13 (14%) targeted therapy, and 57 (63%) chemotherapy or immunotherapy. Among the latter, 38 consented to participate in the study (median age: 75 years). Median cumulative illness rating scale for geriatrics (CIRS-G) was 10 (IQR: 8–12), and median WS 1.09 (IQR: 0.9–1.31) m/s. Older age (*p* = 0.03) and comorbidities (*p* = 0.02) were associated with Grade 3–4 treatment-related adverse events or death within 6 months of accrual. Overall survival was 14.3 (IQR: 6.1-NR) months for patients with WS < 1 m/s versus 17.3 (IQR: 9.2–26.5) for those with WS ≥ 1 m/s (*p* = 0.78). This exploratory study revealed WS to be numerically, yet not significantly, associated with early mortality in older metastatic NSCLC patients. Following these hypothesis-generating results, a larger prospective, multicenter study appears to be required to further investigate this outcome.

## 1. Introduction

Lung cancer is one of the most common cancers in elderly patients, and patients aged 70 years or older represent almost half of those diagnosed with lung cancers [1]. In 2018, lung cancer was the second most frequent cancer diagnosed in the world among females and males aged 65 to 79 years old as those of 80 years or older [2]. At baseline assessment, approximately 60% of patients are diagnosed with inaugural metastases [3]. Lung cancer is still the leading cause of cancer-related death worldwide [3], in spite of revolutionary therapeutic advances made with immunotherapy and targeted therapies over recent decades. Mortality remains high, with 2-year survival rates of 20% for metastatic non-small-cell lung cancer (NSCLC) that is treated using platinum-based chemotherapy [3]. Platinum-based doublet chemotherapy with carboplatin and weekly paclitaxel has become the standard of care for patients over 70 years old with Eastern Cooperative Oncology Group (ECOG) performance status (PS) 0–1; by extension, this standard of care is also applied to selected PS 2 patients, with metastatic NSCLC without addictive mutation, regardless of programmed death-1 (PD-1) ligand 1 (PD-L1) status [4]. In cases with PDL1 staining of more than 50% of cancer cells, pembrolizumab constitutes a valid option for patients in good general condition, in line with the results of the Keynote-024 trial [4,5]. Combination chemotherapy with immunotherapy was not specifically assessed in elderly patients, though nearly 15% of the ≥75-year-olds were accrued into these trials. Overall, elderly patients have habitually been underrepresented in clinical trials [6], as risk factors for early toxicity and mortality have not yet been thoroughly investigated in this patient population. Meanwhile, several causes accounting for the low accrual of elderly patients have been identified, including late diagnosis, comorbidities, treatment abstention, toxicities, as well as co-medications. 

Onco-geriatric tools, including the Comprehensive Geriatric Assessment (CGA), are meant to constitute a global approach designed to help oncologists in their effort to determine which elderly patients could benefit from systemic treatments, without major safety issues, eventually enabling them to adapt the cancer treatment doses [7,8]. This evaluation can prove to be time- and resource-consuming, despite its being the gold standard. Consequently, the challenge is to find an easy and reproducible test, meant to guide the clinician’s decisions. Furthermore, functional and nutritional parameters are likely useful for estimating life expectancy. In the French ELCAPA study, functional parameters, the timed Get Up and Go (GUG) test, as well as malnutrition, were all shown to be able to predict 1-year mortality in older patients undergoing chemotherapy for solid cancers [9]. 

Walking speed (WS) was shown to be a well-established parameter to predict mortality in community-dwelling subjects over 65 years of age [10], including those in healthy condition [11]. WS, independently of GUG, similarly emerged as a predictor of 1-year mortality in elderly patients suffering from solid cancers [12]. Currently, however, data covering cancer patients older than 75 years are still very scarce, while the populations investigated in the studies that address this issue appears to be quite heterogeneous [12]. Besides, of note is that none of these studies have specifically focused on NSCLC patients. 

Older adults are vulnerable to chemotherapy toxicity. However, there are limited data to identify those at risk. In a predictive model consisting of geriatric assessment variables, laboratory test values, as well as patient, tumor, and treatment characteristics, the ability to walk a block was shown to predict toxicity upon chemotherapy with an OR of 1.71 [1.02 to 2.86] [13]. Conversely, to our best knowledge, walking speed as a predictor of toxicity in metastatic NSCLC elderly patients selected for undergoing systemic treatment has never been evaluated.

This study primarily sought to determine whether WS would be a predictor of Grade 3–4 treatment-related adverse events (TRAEs) at 6 months or of early death, in patients ≥70 years of age, with metastatic NSCLC devoid of addictive mutation, while being frontline treated using chemotherapy or immune checkpoint inhibitors (ICI). The secondary objective was to assess the predictive value of other components of geriatric assessment (GA), including sarcopenia, comorbidities, or nutritional assessment.

## 2. Patients and Methods

### 2.1. Study Design and Participants

A prospective observational single-center study was conducted in the thoracic oncology outpatient clinic (Bichat University Hospital, AP-HP Paris, France). Between December 2017 and July 2019, patients who met the following inclusion criteria were recruited: aged 70 years and more, histological or cytological proven NSCLC without oncogenic driver, metastatic or locally-advanced stage without any intent of curative local treatment, systemic treatment-naive, as well as outpatient care setting. Patients were prior to having received their systemic treatment’s first injection, with the decision of systemic treatment already been made by the respective oncologist at that time. Patients treated by best supportive care (BSC) alone were excluded from participating to this study dedicated to patients selected for undergoing first-line systemic treatment. Patients treated by tyrosine kinase inhibitor (TKI) treatment because of addictive oncogenic molecular tumor alterations, were also excluded. Under TKIs indeed, Grade 3–4 TRAEs are known to be uncommon, while Grade 5 are virtually never observed. Moreover, trials dedicated to patients with addictive activating tumor mutation, including elderly people or patients with very poor performance status, have shown high response rates, leading to fast general condition improvement [14].

### 2.2. Data Collection and Baseline Measurements

Demographic, clinical, and specific oncological data were collected, including age, gender, performance status, smoking history, body mass index (BMI), cancer type, cancer stage, treatment type as well as Grade 3–4 TRAE. Geriatric and dietary data were collected, including comorbidities assessed by a board-certified geriatrician (AA, MS, or HE) using cumulative illness rating scale for geriatrics (CIRS-G), mini nutritional assessment (MNA) short form, albuminemia, weight loss within the last 6 months, and 6 m usual WS. Comorbidities, nutritional evaluation, and functional evaluation were collected, based on the International Society of Geriatric Oncology (SIOG) recommendations [7]. CGA was not systematically rated. 

Concerning WS definition, the patient was requested to walk as usual, with technical assistance only provided if needed, with this parameter measured over a short distance such as 4 m (m). Patients had to walk along a corridor with the following indication: “Please begin walking at your normal pace” and following the running order: “Go”. The patients then started walking with 2 m covered before their WS was measured, and they then stopped walking after having walked 2 further meters after the measurement’s ending (8 m in total). WS was measured in m/sec using a chronometer by dividing the distance carried out in meters (4 m) by the time taken in seconds [12]. Different thresholds were proposed in the literature. The frailty threshold was considered to be 1 m/s, whereas the pathological threshold was estimated at 0.8 m/s according to the report by Pamoukdjian et al. [15]. 

Sarcopenia was evaluated based on abdominal computed tomography (CT) scans that were performed as part of routine patient management within 6 weeks prior to inclusion. These CT scans were centrally analyzed by two radiologists who were both blinded to patients’ clinical data (TI; AK) The cross-sectional areas (cm^2^) of the muscles’ sum in the third lumbar spine vertebra (L3) region were computed using Carestream software (Carestream Health, Rochester, NY, USA, Version 12.1.5.1046). Skeletal muscle index (SMI) was calculated as the skeletal muscle area (cm^2^)/height (m^2^) ratio. Sarcopenia was defined as a surface <55.4 cm^2^/m^2^ for men and <38.9 cm^2^/m^2^ for women [16].

### 2.3. Endpoints

Early death and occurrence of Grade 3–4 TRAEs were co-primary study outcomes. TRAEs were graded according to the Common Terminology Criteria for Adverse Events, Version 5.0 (Published: November 27, 2017; U.S. Department of Health and Human Services) and collected during the course of the first-line treatment for a maximum of 6 months after treatment initiation. Early death was defined as occurring within 6 months of treatment initiation. Secondary objectives were to determine the predictive value of other GA components, including sarcopenia, comorbidities, nutritional assessment for early death, or occurrence of Grade 3–4 TRAEs. Finally, another study objective was to test the predictive WS in regard to overall survival (OS). 

In parallel, we assessed the outcome of all consecutive metastatic NSCLC patients, aged 70 years and more, who were diagnosed in our institution during the same period, as identified on pathology files. OS was defined for each patient group, depending on the first-line treatment received (BSC only, TKI, chemotherapy, or immunotherapy). 

### 2.4. Statistical Analysis

This exploratory study, without any assumptions made about results, was conducted to inform the design and recruitment feasibility of a definitive, adequately powered, large-scale multicenter study. As such, it appeared impossible for us to establish a definitive figure for minimal detectable differences. Due to the relatively small patient number, we upfront acknowledged that potentially misleading associations with unidentified variables could be missed.

Quantitative variables were expressed using medians, as well as first and third quartiles [Q1–Q3], and categorical variables using numbers and percentages. In bivariate analysis, quantitative variables were compared based on non-parametric Wilcoxon testing, and categorical variables based on chi-squared or Fisher’s exact tests. Multivariable analysis was not performed due to the limited sample size. 

GA and WS were not performed in all patients that underwent chemotherapy or immunotherapy during the accrual period. We first compared the other baseline characteristics, as well as OS among patients either with or without GA in order to check for selection bias.

Next, we evaluated the characteristics associated with early toxicity and death in participants, but only those with GA. The baseline characteristics were compared according to the incidence of Grade 3–4 TRAEs or death during the first 6 months of treatment. The first day of chemotherapy or immunotherapy was considered as baseline assessment in regard to the 6-month treatment period. 

A sensitive analysis was performed that was designed to compare the proportion of patients receiving a second-line therapy during the follow-up period versus those not receiving such a therapy, according to baseline WS.

Lastly, OS was assessed in all patients that were diagnosed during the inclusion period according to the therapeutic strategy (BSC only, chemotherapy, or immunotherapy), as well as according to WS (< or ≥1 m/s), and treated with chemotherapy or immunotherapy. The date of the biopsy was considered as baseline assessment point to calculate the Kaplan–Meier curves. The status of each patient (alive or deceased) was determined on the study termination date (set to 6 September 2021). No patient was lost to follow-up. The log-rank test was performed to compare survival curves. The Hazard Ratio (HR) with 95% confidence interval (95% CI) was estimated using a Cox model for the association between walking speed and survival. However, due to the limited number of participants, we were not able to perform a multivariable analysis.

Significance of tests was defined by bilateral *p* value < 0.05. Statistical analyses were performed using the JMP^®^ 9.0.3 software (SAS Institute, Cary, NC, USA).

## 3. Results

### 3.1. Patient Baseline Characteristics 

Overall, 145 patients aged 70 and more were treated in our thoracic oncology department between December 2017 and July 2019. Among them, 54 (37%) exhibited localized cancers and 91 (63%) locally-advanced without any intent of curative local treatment or metastatic cancers.

Among the 91 patients with locally-advanced or metastatic cancer, 21 (23%) received BSC, 13 (14%) displayed a driver mutation enabling them to benefit from oral targeted therapy, and 57 (63%) were offered chemotherapy or immunotherapy. Of the latter 57 patients, 38 (67%) agreed to participate to our study and underwent a GA. The other 19 patients (33%) were not accrued, as two patients refused to participate, one was not included because of an associated hepato-carcinoma, and the others did not complete the GA for various reasons (Figure 1). 

Baseline characteristics from patients included in the study (*n* = 38) did not differ from those that did undergo GA (*n* = 19) (Appendix A). Moreover, median OS was not statistically different between patients included or not included into the study (16.0 [IQR: 8.7-NR] vs. 12.8 [IQR: 5.6–20] months, respectively, *p* = 0.28).

The 38 included patients displayed a median age of 75 years (IQR, 71.8–78.5), a large majority of patients being male (84%) (Table 1). Overall, 90% were smokers, most of them having stopped smoking before GA was performed. Considering tumor histology, 76% of cases were non-squamous carcinoma and 24% were squamous cell carcinoma. One-third of patients exhibited Stage III tumor, their tumor volume being too large for ‘curative’ intent radiation therapy, while two-thirds of patients suffered from Stage IV cancer. Overall, 22 patients (58%) were PS 0–1 and 16 (42%) PS 2. The median CIRS-G was 10 [IQR, 8–12]. Overall, 10 patients (26%) were treated using immunotherapy, and 28 (74%) by means of chemotherapy.

Concerning nutritional status, 10 patients (26%) had lost ≥3 kg in the previous month, nine (23.7%) were considered as malnourished, while 20 (52.6%) were at risk of malnutrition according to the MNA. Overall, 32 patients (85%) displayed a low skeletal muscle index, which is an indicator of sarcopenia. Median albuminemia was 39 (IQR: 33–43) g/L. 

The median WS was 1.09 (IQR: 0.90–1.31) m/s, with 13 patients (34%) having a WS < 1 m/s, and six (16%) a WS < 0.8 m/s. 

On their initial treatment, eleven patients (29%) experienced a Grade 3–4 TRAE or died within 6 months of inclusion (Table 2). Older age (*p* = 0.03) and CIRS-G (*p* = 0.02) were associated with Grade 3–4 TRAEs or death at 6 months. For the chemotherapy-treated patient subgroup (*n* = 28), loss of appetite (*p* = 0.02) and CIRS-G (*p* = 0.03) were both significantly associated with Grade 3–4 TRAEs or early death. 

WS did not predict occurrence of Grade 3–4 TRAE or early death, either for the total population or for the chemotherapy-treated patient subgroup (*p* = 0.5). 

### 3.2. Overall Survival 

The median follow-up duration for the 91 metastatic cancer patients was 12.4 (IQR: 5.6–21.3) months, the maximum follow-up duration being 49.3 months. Median OS for these patients was 12.6 (IQR: 5.6–25.5) months. OS was different according to therapeutic strategy (*p* < 0.001) (Figure 2). OS was 15.2 (IQR: 11.3–20) months for the 57 chemotherapy- or immunotherapy-treated patients, and 2.2 (IQR, 0.9–6.2) months for those receiving BSC alone (*n* = 21), with a median follow-up of 15.9 (IQR: 7–25) months. 

In the 38 participants with GA, WS was not associated with OS. Kaplan–Meier curves according to WS are displayed in Figure 3. OS was 14.3 (IQR: 6.1-NR) months for patients with WS < 1 m/s, and 17.3 (IQR: 9.2–26.5) months for patients with WS ≥ 1 m/s (*p* = 0.78), with a median follow-up time of 15.6 (IQR: 8.5–29.6) months. The estimated percentage of patients who were alive at 24 months was 44% in the WS < 1 m/s patient group and 30.8% in the WS ≥ 1 m/s group (*p* = 0.42). 

In the WS ≥ 1 m/s group, 19/25 (76%) patients received second-line therapy versus 4/13 (30.8%) in the WS < 1 m/s group (*p* = 0.018). The other patients died upon first-line therapy or received only BSC at disease progression. 

## 4. Discussion

In our study, WS assessment performed just prior to systemic treatment initiation in a population of Stage IIIB–IIIC or IV NSCLC patients, aged 70 years or more, who were already selected by the oncologist to undergo chemotherapy or immunotherapy, was not a significant predictor of early death or Grade 3–4 TRAEs. Nevertheless, according to our analysis, higher age (*p* = 0.03) and comorbidities based on CIRS-G (*p* = 0.02) were both predictive of adverse outcomes. Loss of appetite was also predictive of poor OS and toxicity, yet exclusively in the chemotherapy group (*p* = 0.02). Analysis of NSCLC histological subtype was not performed because of the low numbers in subset categories and thus the low probability that such variable could have significantly influenced survival analysis.

In spite of a potential imbalance in second-line treatments received, these were unlikely to have influenced OS, if we consider that median first-line progression-free survival (PFS) with chemotherapy usually amounts to 3–4 months, and PFS with immunotherapy to 6–7 months, with the latter depending on PD-L1 expression [17]. 

In previous studies involving older cancer patients, WS was shown to be likely correlated with life expectancy, frailty, and early mortality [12,15,18,19]. Specifically, a WS threshold of 1 m/s was more commonly associated with the presence of at least one abnormality on the geriatrician’s EGA in an onco-geriatric population, as shown in the study by Pamoukdjian et al. [15]. It is impossible to exclude that the lack of association between WS and early death or Grade 3–4 TRAE, as observed in our population, could be accounted for by the study’s small sample size. We did not find any impact of comorbidities, age, or nutrition status either, which suggests that besides a lack of power, our study cohort may actually differ from that of previous studies. Indeed, we have focused on lung cancer patients, representing a patient group with specific tobacco-induced comorbidities and aging characteristics. This could, in fact, have concealed other prognostic variables. Another plausible explanation could be that patients were assessed on the day of their first perfusion of chemo- or immunotherapy, after having already been selected by the oncologist to undergo systemic treatment. As a result, patients not eligible for systemic treatment prior to GA assessment, as estimated by the oncologist, who were thus receiving BSC alone, were not included in our analysis. Given this context, the population of interest in our study was likely to differ from that of previously published studies. 

In the first study published by Pamoukdjian et al. [12], involving various cancer sites, as well as either localized or metastatic tumor stages, WS was correlated with early mortality. In this study, median age was 80.6 years, whereas it was only 75 years in our study. In addition, Pamoukdjian’s population may have differed from ours due to their assessments being made prior to therapeutic decision-making. Patients who subsequently received BSC alone were also accrued in these studies, as opposed to our study, in which patients were already selected by the oncologist to underdo chemo- or immunotherapy. Half of Pamoukdjian’s population displayed a WS < 0.8 m/s, with a median WS of 0.79 m/s. Yet, in our study, only 16% of patients displayed a WS < 0.8 m/s, with a median WS of 1.09 m/s. These figures clearly demonstrate a certain level of physical performance, which actually suggests our population to be already highly selected.

It is interesting to remind the reader that Couderc’s study [20], which involved thoracic cancers (NSCLC, small-cell lung cancer (SCLC), and mesothelioma) at all stages (45% with Stage IV disease), also did not find any association between WS and OS. Conversely, the Timed Up and Go (TUG) test, one leg balance test for the patients under 79 years 80 s, and handgrip test for the patients 80 years or more were all associated with OS. With regard to this latter study, it would have been interesting to apply the Short Physical Performance Battery (SPPB) to this study population so as to determine whether the association of these tests, including chair lifts, balance, and WS, would have been more effective than using WS alone. 

CGA has been applied in a randomized Phase 3 trial, ESOGIA, which focused on metastatic NSCLC patients and compared decision to treat based on age and either PS or GA [21]. In that well-designed study, adaptation of treatment according to CGA, as performed by an oncologist, failed to demonstrate survival benefits, suggesting that these scores could be prognostic in nature rather than predictive. However, comorbidities (Charlson Comorbidity Index ≥ 2), geriatric syndrome, and nutrition as reflected by BMI were parameters influencing Treatment Failure Free Survival (TFFS). In our study, for patients undergoing chemotherapy, one of the nutritional criteria (loss of appetite) turned out to be similarly significant (*p* = 0.02), supporting once more that our negative results were not uniquely linked to insufficient study power. 

In the IFCT-0501 randomized Phase 3 trial [22], which established the standard of care for elderly people with metastatic NSCLC and overall accrued 451 patients, an association was found between weight loss before randomization and survival, whereas scores of Activities of Daily Living (ADL) and Mini-Mental State Examination (MMSE) failed to specifically predict survival in patients accrued into the experimental arm (platinum-based chemotherapy), as compared with standard single-agent arm. Age was also not correlated with survival, contrary to what we found. However, it is highly probable that octogenarians included into a Phase 3 trial were more drastically selected than those participating to an observational study in a real-life setting. 

When we look at the population treated by BSC in our study, OS was 2.2 (IQR: 0.9–7.5) months. This figure is close to the OS observed in exclusively BSC-treated patients from the ESOGIA study, herein amounting to 2.8 months [21]. Nevertheless, this population has been clearly heterogeneous in our study, given that 30% of these patients survived beyond 6 months. This proportion of patients, however, may have been under-treated, since they had no access to a specific treatment. Indeed, it is probably the specific population for which a comprehensive standardized geriatric pre-treatment assessment would have been useful to both the oncologists and patients in order to select their optimal disease management. It would have been interesting to understand whether WS would have improved the selection of BSC-treated patients designed to identify the 30% that are still alive at 6 months. Finally, in our study conducted from 2017 to 2019, OS was 15.2 (IQR: 11.3–20) months for patients treated by means of chemotherapy or immunotherapy in first-line setting. As a reminder, OS was 10 months for patients treated using doublet chemotherapy in the CGA arm of the ESOGIA trial performed between 2010 and 2012, and 10.3 months for patients treated using doublet chemotherapy in the IFCT-0501 trial performed between 2006 and 2009. This improvement in patient survival can likely be accounted for by the approval of immunotherapy in 2016 and 2017, in second-line and first-line settings, respectively, in France. In addition, good results obtained with associations of chemotherapy and immunotherapy, or double immunotherapy, could soon render it necessary to reconsider the standard of treatments for older patients with advanced NSCLC.

To our knowledge, there is no other prospective study reported in the literature that has so far investigated WS as a predictor of toxicity and early mortality prior to establishing treatment of metastatic lung cancer in elderly patients. This study does, however, have some limitations. First, it was a single-center survey. Second, this was an exploratory study, and its relatively small patient numbers could have missed potentially essential prognostic associations.

## 5. Conclusions

In this exploratory study, WS was not revealed to be predictive of early mortality or Grade 3–4 TRAEs in a population of average 75-year-olds, already selected by the oncologist to receive chemotherapy or immunotherapy in the first-line setting for metastatic NSCLC. Comorbidities and age were found to be predictive of survival and treatment toxicity, so was loss of appetite, yet the latter only for patients treated using chemotherapy. We failed to demonstrate that WS, assessed on the day of the first chemotherapy or immunotherapy perfusion, could be a significant criterion in the pre-treatment phase, as an aid to the therapeutic choice. Further research specifically devoted to elderly patients is thus urgently required. Such research would need to determine reliable clinical markers that should help us identify those patients that are more likely to benefit from therapy. Indeed, this research appears mandatory so as to enable us to draw definitive conclusions concerning this patient population.

## Figures and Tables

**Figure 1 cancers-14-01344-f001:**
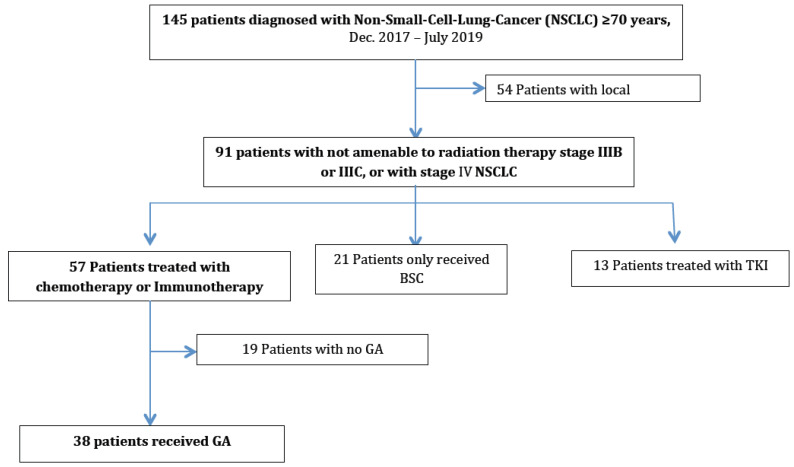
Flow-chart. GA: geriatric assessment; TKI: tyrosine kinase inhibitor; BSC: best supportive care.

**Figure 2 cancers-14-01344-f002:**
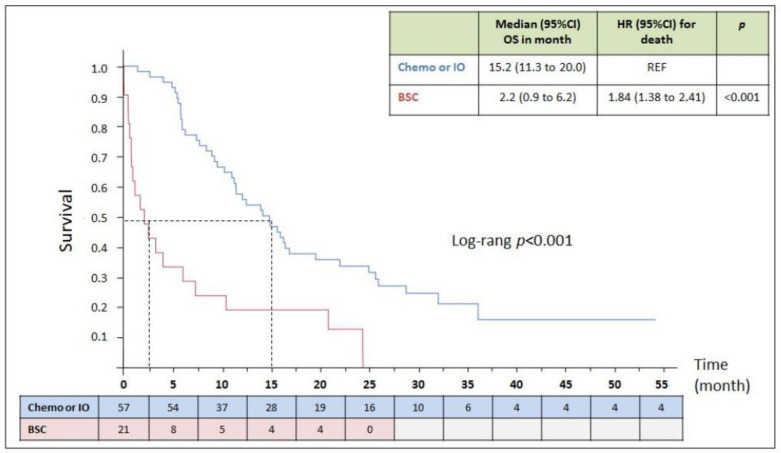
Overall survival according to therapeutic strategy (*n*= 78). The Kaplan–Meier curves represent the survival over time in patients diagnosed during the inclusion period according to the therapeutic strategy: Chemotherapy or Immunotherapy, or BSC. The day of the biopsy was considered as baseline. The number of participants alive over time is detailed in the table under the curves. Chemo: chemotherapy; IO: immunotherapy; BSC: best supportive care.

**Figure 3 cancers-14-01344-f003:**
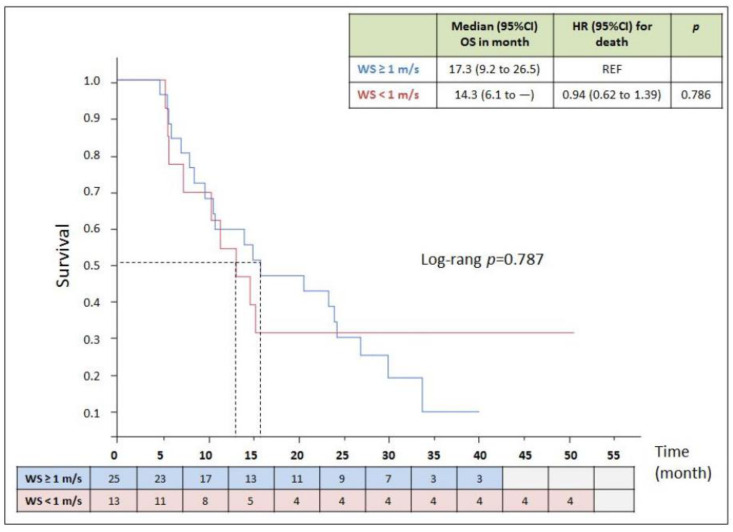
Overall survival according to walking speed (*n* = 38) The Kaplan–Meier curves represent the survival over time according to the walking speed (< or ≥1 m/s) in participants treated with chemotherapy or immunotherapy. The day of the biopsy was considered as baseline.

**Table 1 cancers-14-01344-t001:** Patient characteristics at inclusion.

Characteristics	Overall Patient Population	Patients with Chemotherapy	Patients with Immunotherapy
	*n* = 38	*n* = 28	*n* = 10
Age, years (IQR)	75 (71.8–78.5)	75 (71.3–79.5)	76 (71.8–79.5)
Male gender, *n* (%)	32 (84)	23 (82)	9 (90)
Performance Status			
0–1, *n* (%)	22 (58)	16 (57)	6 (60)
2, *n* (%)	16 (42)	12 (43)	4 (40)
Smokers, *n* (%)	34 (90)	25 (89)	9 (90)
Pack-years (IQR)	50 (20–80)	50 (19–80)	80 (30–120)
Histology			
Squamous, *n* (%)	9 (24)	5 (18)	4 (40)
Non-squamous, *n* (%)	29 (76)	23 (82)	6 (60)
Stage			
Stage III, *n* (%)	13 (34)	10 (36)	3 (30)
Stage IV, *n* (%)	25 (66)	18 (64)	7 (70)
Weight, Kg (IQR)	75 (65.8–85)	76 (68.8–85)	72 (49.8–83)
Body mass index, Kg/m^2^ (IQR)	26.6 (23.5–29.5)	27.1 (24.1–29.4)	25.7 (18.4–29.9)
Body surface, m^2^ (IQR)	1.90 (1.7–2)	1.90 (1.8–2)	1.85 (1.31–1.95)
Weight loss >3 kg in 1 months, *n* (%)	10 (28)	7 (25)	3 (38)
Loss of appetite, *n* (%)	15 (39)	10 (36)	5 (50)
Plasma Albumin, g/L (IQR)	39 (33–43)	37.5 (32.3–43)	40 (34.5–41.8)
MNA-SF score/14 (IQR)	10 (8–11)	11 (9–12)	8 (5–11)
MNA-SF categories			
Normal nutritional status (12–14 points), *n* (%)	9 (24)	7 (25)	2 (20)
Risk of malnutrition (8–11 points), *n* (%)	20 (52)	17 (61)	3 (30)
Malnutrition (0–7 points), *n* (%)	9 (24)	4 (14)	5 (50)
CIRS-G, *n* (%)	10 (8–12)	9 (7–12)	11 (8–13)
Skeletal muscle mass index (SMI), cm^2^/m^2^ (IQR)	47.3 (40.5–52.2)	48.9 (39.3–52.5)	43.1 (40.5–49.5)
Sarcopenia according to SMI *, *n* (%)	32 (84)	24 (86)	8 (80)
Walking speed, m/s (IQR)	1.09 (0.9–1.31)	1.13 (0.91–1.29)	1.06 (0.66–1.35)
Walking Speed < 1 m/s, *n* (%)	13 (34)	10 (36)	3 (30)

Quantitative variables were expressed using the median value and interquartile range (IQR), and categorical variables were given using the number and percentage, *n* (%). * Sarcopenia was defined as an area <55.4 cm²/m² for men and <38.9 cm²/m² for women. CIRS-G: Cumulative Illness Rating Scale-Geriatric; MNA-SF: Mini Nutritional Assessment-Short Form; SMI: Skeletal muscle mass index.

**Table 2 cancers-14-01344-t002:** Characteristics at inclusion according to occurrence of Grade 3–4 TRAEs or early death.

Characteristics	Outcomes during the First 6 Months of Treatment				
	No *n* = 27	Yes *n* = 11			*p*
		Grade 3–4 TRAE or Death *n* = 11	Alive with Grade 3–4 TRAE *n* = 5	Deceased *n* = 6	
Age, years (IQR)	74 (71–78)	76 (75–82)	80 (74–81)	76 (75–80)	0.034
Male gender, *n* (%)	22 (81)	10 (91)	5 (100)	5 (83)	0.650
Performance status					
0–1, *n* (%)	16 (59)	6 (55)	3 (60)	3 (50)	0.790
2, *n* (%)	11 (41)	5 (45)	2 (40)	3 (50)	
Pack-years (IQR)	50 (25–80)	65 (28–105)	40 (25–100)	65 (25–105)	0.690
Histology					
Squamous, *n* (%)	8 (30)	1 (9)	0 (0)	1 (17)	
Non-Squamous, *n* (%)	19 (70)	10 (91)	5 (100)	5 (83)	0.237
Stage					
Stage III, *n* (%)	10 (37)	3 (27)	0 (0)	3 (50)	0.714
Stage IV, *n* (%)	17 (63)	8 (73)	5 (100)	3 (50)	
>Weight, Kg (IQR)	74 (61–85)	75 (73–80)	75 (75–78)	79 (67–90)	0.675
Body mass index, Kg/m^2^ (IQR)	25.6 (21–29.7)	27.5 (26.2–29.4)	27.5 (27.1–28.8)	28.1 (23.4–31.2)	0.253
Body surface, m^2^(IQR)	1.90 (1.70–2.00)	1.90 (1.80–1.90)	1.90 (1.90–1.90)	1.90 (1.70–2.13)	0.791
Weight loss > 3 kg in 1 month, *n* (%)	6 (24)	4 (36)	2 (40)	2 (33)	0.446
Loss of appetite, *n* (%)	8 (30)	7 (64)	4 (80)	3 (50)	0.052
Plasma albumin, g/L (IQR)	40 (31–43)	37 (33–41)	33 (31–39)	39 (33–45)	0.699
MNA-SF (/14), *n* (IQR)	10 (8–12)	10 (7–11)	11 (7–11)	9 (7–11)	0.361
CIRS-G (IQR)	8 (7–11)	12 (10–15)	10 (8–12)	13 (11–16)	0.020
Skeletal muscle mass index, cm^2^/m^2^ (IQR)	45.1 (38.8–51)	52.0 (42.6–53.3)	52.0 (46.8–56.1)	47.3 (36.6–53.4)	0.227
Walking speed, m/s (IQR)	1.10 (0.95–1.34)	1.07 (0.68–1.25)	1.15 (0.83–1.39)	0.99 (0.63–1.18)	0.562
Walking speed < 1 m/s, *n* (%)	9 (33)	4 (36)	1 (20)	3 (50)	0.858

Quantitative variables were expressed using the median value and interquartile range (IQR), and categorical variables were given using the number and percentage, *n* (%). The *p*-values are given for the comparison of participants with (*n* = 11) or without (*n* = 27) outcomes (Grade 3–4 TRAE or death) during the first 6 months of treatment. Quantitative variables were compared using the Wilcoxon test and categorical variables using the chi-squared or Fisher’s tests. CIRS-G: Cumulative Illness Rating Scale-Geriatric; MNA-SF: Mini Nutritional Assessment-Short Form; PY: Pack-Years; TRAE Treatment-Related Adverse Event.

## Data Availability

The data presented in this study are available on request from the corresponding author.

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
