# Peer review of "Measuring Walking Speed Failed to Predict Early Death and Toxicity in Elderly Patients with Metastatic Non-Small-Cell Lung Cancer (NSCLC) Selected for Undergoing First-Line Systemic Treatment: An Observational Exploratory Study"

_cancers, 2022, doi:10.3390/cancers14051344_

Round 1
Reviewer 1 Report
I’m thanking the authors for amending the manuscript and I understand their limitations for recruiting more cases. I believe this is clearly explained and I’m looking forward reading a new paper from them including larger series of cases.
I just have a minor comment for the new version of the manuscript, and it is related to the tilte.
In English, the word “determination” means “the ability to keep doing something difficult”. I know that is different in French, where “dètermination” is defined as “Action de déterminer quelque chose, de le définir avec précision”. Thus, your title for an English speaker sounds like “Walking speed persistence (tenacity, perseverance) failed to predict early death…” May I suggest rewording like: “Measuring walking speed fails to predict …”
Author Response
We very much thank again the Reviewer 1. As suggested, the new proposed title will be therefore:
“Measuring walking speed fails to predict early death and toxicity in elderly patients with metastatic non-small-cell lung cancer (NSCLC) selected for undergoing first-line systemic treatment: an observational exploratory study.”

Reviewer 2 Report
Dear authors,
Your exploratory analysis gives interesting information on how walking speed could represent a prediction of early death and toxicities in patients with advanced/metastatic NSCLC.
I have some comments:
As for introduction:
- You repeat that “lung cancer is common in elderly adults” both in the summary and in the introduction. Maybe you mean that lung cancer is one of the most common cancer in elderly patients? Otherwise, specify the incidence for elderly adults in France or in the word in the last year.
- In the study rational, you stated “Walking speed (WS) was shown to be a well-established parameter to predict mortality in community-dwelling subjects over 65 years of age”. There is some rational / previous studies about correlation between WS and treatment related adverse events? It is not clear in the introduction section.
As for the methods:
- There is some rational to eliminate TKIs inhibitors patients? These drugs were also used to treat advanced/metastatic NSCLC. Please clarify this point.
- Line 106: delete “were proposed”. You mean you excluded these patients right?
- Specify the time window in which the treatment related AEs were searched (line 147).
- It is unclear what is the index date. Biopsy? Diagnosis? Start of first-line therapy?
- Line 180: Before you say that you included only those patients receiving chemo o immunotherapy and those receiving GA. Why do you want to look at the survival of patients that you previously excluded? I would recommend you to be consistent with your study objectives.
- Line 186: You compare the survival curves by using log rank test. Why did you not perform a multivariate COX analysis? I think this strategy should be more appropriate in observational studies.
As for the results:
- Figure 1: You used * that were not reported in the figure caption.
- Line 204: did you mean “that did not undergo GA”?
- Table 2. Which are the “yes” patients? Those who had at least one event? I suggest to report other two columns in the table. The first two for those who died in the 6 months, and the other two for those who experienced a TRAEs.
- Line 252: I think the analysis of the survival between chemo/immune vs BSC is out of the scope of this work. Moreover chemo e immunotherapy are very different drugs. Why did you put them together? In the text you also reported OS for TKIs users that you say you excluded from the analysis in the method section (these were also not described in the tables). Is it right? I suggest to remove this analysis from the manuscript and suggest you to be consistent with the study objective.
- Figure 3. Is it possible to compare survival using a COX (see comment in the method section)? I think the main hurdle to study survival is the few patients included. Anyway these patients could have a different profiles. Have you looked at different NSCLC histologies (PMID: 34885238, PMID: 21371774)? If not please discuss in the discussion section (i.e. information non available).
As for discussion:
- Do you have any suggestions for the conduction of larger observational studies? Where information on WS or GA could be found (PMID: 34300130)?
Round 2
Reviewer 2 Report
Dear authors,
Thank you for your reply, which overall responds to my questions.
This manuscript is a resubmission of an earlier submission. The following is a list of the peer review reports and author responses from that submission.
Round 1
Reviewer 1 Report
I have a few comments on this manuscript and I thank the authors for reading and considering them.
- As discussed by the authors, with such a small number of cases, hidden interactions between variables in the analysis are hard to be unveiled by logistic regression analysis. I agree with the authors on the need of increasing numbers recruiting patients from other institutions.
- The title is leading the reader to expect WS having some predictive value but not negative findings. I’m suggesting this title instead: “Usual walking speed does not predict early death and toxicity, etc…”
- In the Conclusions section, lines 372-373 are redundant and simply repeating some of the statements above in the same section.
- In the same section, lines 373 (from “Good…”) to 376 (up to “…NSCLC”) cannot be concluded from the data presented in the results section. These statements are good to be included in the discussion section but not here.